# MC-ASFF-ShipYOLO: Improved Algorithm for Small-Target and Multi-Scale Ship Detection for Synthetic Aperture Radar (SAR) Images

**DOI:** 10.3390/s25092940

**Published:** 2025-05-07

**Authors:** Yubin Xu, Haiyan Pan, Lingqun Wang, Ran Zou

**Affiliations:** School of Information Science, Shanghai Ocean University, Shanghai 201306, China; 2226234@st.shou.edu.cn (Y.X.); hy-pan@shou.edu.cn (H.P.); 2152730@st.shou.edu.cn (R.Z.)

**Keywords:** ship detection, deep learning, multi-scale feature fusion, small object detection, remote sensing SAR image, YOLO11

## Abstract

Synthetic aperture radar (SAR) ship detection holds significant application value in maritime monitoring, marine traffic management, and safety maintenance. Despite remarkable advances in deep-learning-based detection methods, performance remains constrained by the vast size differences between ships, limited feature information of small targets, and complex environmental interference in SAR imagery. Although many studies have separately tackled small target identification and multi-scale detection in SAR imagery, integrated approaches that jointly address both challenges within a unified framework for SAR ship detection are still relatively scarce. This study presents MC-ASFF-ShipYOLO (Monte Carlo Attention—Adaptively Spatial Feature Fusion—ShipYOLO), a novel framework addressing both small target recognition and multi-scale ship detection challenges. Two key innovations distinguish our approach: (1) We introduce a Monte Carlo Attention (MCAttn) module into the backbone network that employs random sampling pooling operations to generate attention maps for feature map weighting, enhancing focus on small targets and improving their detection performance. (2) We add Adaptively Spatial Feature Fusion (ASFF) modules to the detection head that adaptively learn spatial fusion weights across feature layers and perform dynamic feature fusion, ensuring consistent ship representations across scales and mitigating feature conflicts, thereby enhancing multi-scale detection capability. Experiments are conducted on a newly constructed dataset combining HRSID and SSDD. Ablation experiment results demonstrate that, compared to the baseline, MC-ASFF-ShipYOLO achieves improvements of 1.39% in precision, 2.63% in recall, 2.28% in 
${AP}_{50}$
, and 3.04% in 
AP
, indicating a significant enhancement in overall detection performance. Furthermore, comparative experiments show that our method outperforms mainstream models. Even under high-confidence thresholds, MC-ASFF-ShipYOLO is capable of predicting more high-quality detection boxes, offering a valuable solution for advancing SAR ship detection technology.

## 1. Introduction

As an important task of maritime surveillance, ship detection plays an irreplaceable role in various fields such as marine economic development, search and rescue operations, maritime transportation, military reconnaissance, and security law enforcement [1,2,3,4]. Marine ship detection frequently relies on remote sensing imagery due to its comprehensive spatial coverage, economic efficiency, and rapid data availability [5]. This includes synthetic aperture radar (SAR), which is the preferred data source for ship detection, largely owing to its all-weather and all-day imaging capabilities [6,7,8,9].

Traditional SAR ship detection methods (such as CFAR) rely on manual parameter tuning and exhibit significant performance degradation in complex sea conditions or dense ship scenarios [10,11,12,13], which has gradually shifted the research focus towards more robust detection algorithms. In recent years, artificial intelligence technologies, especially CNNs, have brought breakthrough progress to ship detection due to their powerful nonlinear feature extraction and efficient feature representation capabilities [14]. Using the SSDD dataset as an example, the 
${mAP}_{50}$
 for ship detection has improved from 78.8% to 97.8% [15]. Current deep-learning solutions for object detection adopt either two-stage or single-stage detectors. Two-stage algorithms primarily comprise two fundamental processes: (1) generation of region proposals and (2) classification of features and bounding box regression within the proposed regions [16]. Representative algorithms consist of the R-CNN series [17,18,19]. Ke et al. [20] introduced deformable convolution kernels to improve Faster R-CNN, enhancing the model’s adaptability to geometric deformations of ships, which increased the 
mAP
 by 2.02%. Zhao et al. [21] proposed a two-stage detector named the Attention Receptive Field Pyramid Network (ARPN), which enhances non-local feature associations and refines multi-level representations, thereby boosting the performance of multi-scale ship detection in SAR imagery. While two-stage object detection algorithms achieve higher detection accuracy, they entail greater computational complexity and resource consumption. Single-stage algorithms (such as SSD, RetinaNet, FCOS, and the YOLO series) eliminate the need for region proposal generation and can directly predict bounding boxes and class probabilities through anchor boxes. Wang et al. [22] optimized SAR ship detection by combining SSD with data augmentation and transfer learning, effectively reducing false detection rates and achieving higher target localization precision. Miao et al. [23] proposed an improved lightweight RetinaNet that reduces parameter count and computational overhead without compromising detection accuracy. Zhu et al. [24] presented an enhanced FCOS + ATSS network that achieved an 8.3% improvement in 
AP
 compared to the baseline model. The YOLO series dominates ship detection as the most popular single-stage approach [16]. Simultaneously, Transformer models based on self-attention mechanisms effectively capture global relationships between image pixels, demonstrating tremendous potential in ship detection. For instance, the SMEP-DETR model proposed by Yu et al. [25] effectively suppresses speckle noise in SAR images while enhancing edge features, enabling small target ships to maintain high localization accuracy even in complex background environments. The RDB-DINO approach introduced by Qin et al. [26] mitigates confusion between small ships and complex backgrounds while enhancing the feature representation of small ships, significantly improving detection performance for small ships. Transformers possess superior global modeling capabilities [27] and outperform certain single-stage detection models in object detection tasks.

Despite significant advances in deep-learning-based SAR ship detection methods, multiple challenges persist: ship types are increasingly diverse with substantial size variations, and ship pixel areas within the same dataset can differ by nearly 1000-fold [28]. CNNs with fixed receptive fields struggle to adapt to these multi-scale characteristics. Additionally, ships may appear in complex scenarios including near-shore ports, open ocean, coastlines, and inland waterways, making them prone to background confusion [2,29]. Furthermore, densely distributed fleets further increase detection difficulty [30]. Moreover, SAR ship datasets contain a high proportion of small targets [31], which provide limited feature information [32,33] and are susceptible to background interference, severely constraining detection performance.

To address these challenges, researchers have explored various improvement strategies. Generally, these methods primarily include multi-scale feature fusion optimization, attention mechanisms, enhanced feature extraction networks, and loss function improvements. Regarding multi-scale feature fusion, Liu et al. [34] combined a Feature Pyramid Network (FPN) with Scale-Equalizing Pyramid Convolution (SEPC) to enhance YOLOv4’s multi-scale feature processing capability. Chen et al. [35] innovatively applied a k-means clustering algorithm based on Shape Similarity Distance (SSD) metrics to optimize FPN, effectively resolving small ship detection problems in complex environments. Li et al. [36] proposed a Balanced Shifting Multi-scale Fusion (BSMF) module based on YOLOv8, significantly improving detection performance for targets of different scales. In terms of attention mechanism applications, Cui et al. [37] embedded a Convolutional Block Attention Module (CBAM) into the feature map cascade process of the pyramid network, connecting layers sequentially from top to bottom to extract rich multi-resolution semantic features and highlighting salient features. Yang et al. [38] introduced a Coordinate Attention Module (CoAM) in single-stage detection networks to mitigate complex background interference and enhance semantic feature representation. Zhou et al. [39] proposed an attention aggregation network WEF-Net that coordinates semantic information across feature layers of different resolutions, simultaneously enhancing multi-scale detection capabilities and background suppression effects. Tang et al. [40] integrated the BiFormer attention mechanism into YOLOv7-tiny, significantly reducing false positives and false negatives in near-shore scenarios. Wang et al. [41] proposed a lightweight multi-scale SAR ship detection model called MSSD-Net, which introduces the Multi-Scale Coordinate Attention Module (MSCA) to effectively capture global information from input feature maps and enhance the capability to process features across different scales. Regarding feature extraction network enhancements, Fan et al. [42] proposed the CSDP module, which employs deep large-kernel convolutions to enlarge the receptive field of shallow layers, thereby enhancing the representation of small target features. Yang et al. [38] proposed a Receptive Field Improvement Module (RFIM) to enhance detection capabilities for ships of various scales. In terms of loss function improvements, Li et al. [36] introduced Gaussian Wasserstein distance loss, while Fan et al. [42] proposed the MPDIOU loss function, effectively alleviating the class imbalance problem between small SAR ships and backgrounds.

Despite these improvements enhancing SAR ship detection performance, research has predominantly focused on enhancing individual detection models that address either small target detection challenges or multi-scale ship detection issues individually. Research that simultaneously addresses both challenges within a unified detection framework is still relatively limited. This study aims to propose an improved ship detection model that, on the one hand, enhances small target feature representation and distinctiveness by increasing attention allocation to small targets, reducing information loss in high-level feature representations, and strengthening the model’s perception of small targets and their contextual background relationships. On the other hand, it resolves cross-scale feature conflict problems inherent in traditional FPN structures, thereby enhancing feature consistency across targets of different scales. This dual improvement strategy significantly enhances model performance in small target recognition and multi-scale target detection, providing a more effective solution for ship detection in SAR images.

To overcome these limitations, we propose MC-ASFF-ShipYOLO (Monte Carlo Attention—Adaptively Spatial Feature Fusion—ShipYOLO), an improved single-stage detector based on YOLO11 [43,44]. Our main contributions are as follows.

We introduce the MCAttn module to enhance backbone network performance. This module randomly selects one attention map from three different scales through stochastic sampling pooling operations for the weighting of feature maps. By capturing multi-scale information, the MCAttn module improves the backbone network’s ability to discriminate small ship morphology and position, enhances focus on small targets, prevents information loss during network deepening, and strengthens contextual relationship learning. The module effectively enhances the feature representation of small ships, thereby improving the model’s detection capability for small ship targets.We incorporated the ASFF module into the detection head, which adaptively learns spatial fusion weights across multi-scale feature layers and performs dynamic feature integration. This approach ensures the consistency of ship feature representations across different scales, effectively mitigating the problem of feature conflicts between varying feature layers. Consequently, the model’s multi-scale ship detection capability is enhanced.We propose MC-ASFF-ShipYOLO, an improved deep-learning ship detection model that demonstrates superior precision compared to other baseline models when evaluated on the mixed HRSID and SSDD datasets. Our detection framework effectively improves the critical challenges arising from small target recognition and multi-scale feature representation in SAR ship detection tasks. The experimental results validate the efficacy of our approach and provide valuable methodological guidance for future research in maritime target detection.

The subsequent sections are arranged accordingly: Section 2 introduces the newly constructed hybrid SAR ship detection dataset and elaborates on the proposed framework and improvement modules. Section 3 presents comparative experiments and ship detection results. Section 4 presents experimental results and comprehensively analyzes the contributions of the improvement modules. Finally, Section 5 provides a conclusion of the entire paper.

## 2. Materials and Methods

This section presents the experimental datasets, data preprocessing methods, implementation details, improved methods, and evaluation metrics.

### 2.1. Dataset Introduction

This study employs a hybrid dataset, constructed from HRSID and SSDD, as the basis for model training and performance evaluation. Details of the hybrid dataset can be found in the Appendix A. Released in 2020, HRSID [45] is a high-resolution SAR image dataset designed for ship detection and segmentation tasks. It comprises 5604 images with a resolution of 800 × 800 pixels, ranging from 0.5 m to 3 m, and includes diverse scenes such as open seas and coastal ports. SSDD utilizes the improved version officially released by Zhang et al. [13] in 2021, comprising 1160 images with resolutions varying from 1 m to 15 m and inconsistent image dimensions (height 190–526 pixels, width 214–668 pixels). This dataset provides three annotation methods: (1) BBox; (2) RBox; (3) PSeg. To unify model input, the HRSID dataset employs axis-aligned vertical bounding boxes (BBoxes) as its annotation format; for the SSDD dataset, we also use the BBox annotation format. Additionally, we apply zero-padding to the SSDD dataset to achieve uniform 800 × 800 dimensions. This approach preserves original image information without distortion while simultaneously unifying input dimensions, improving model compatibility and computational efficiency.

After processing, we constructed a hybrid experimental dataset containing 6764 SAR images. An 8:2 ratio was used to divide the training and validation sets (5411 images for training, 1353 images for validation), ensuring experimental reliability and model generalization capability. Following the target size definition standards of the MS COCO dataset, in 800 × 800 SAR images, we categorize ship targets into three classes: large targets (area > 1%, pixel range 
(80×80, 800×800])
, medium targets (0.5% < area ≤ 1%, pixel range 
(40×40, 80×80])
, small targets (0 < area ≤ 0.5%, pixel range 
(0, 40×40])
.

Figure 1 displays complex background environments, densely distributed small ships, and example scenes of multi-scale ship targets. Figure 2 presents the quantity distribution and proportion of ship targets at various scales in both training and validation sets, with data showing that small ships account for as much as 91.3% and 91.5% in the training and validation sets, respectively. This highlights the primary challenges faced in SAR ship detection. Therefore, improving models to suppress noisy background interference, enhance multi-scale detection capability, and increase small target feature representation and attention is key to improving overall model detection performance.

### 2.2. Implementation Details

In this study, all experiments were conducted using PyTorch on a PC with Intel(R) Xeon(R) Gold 6430 CPU @ 2.10 GHz and NVIDIA GeForce RTX 4090 (24,564 MiB) GPU. The PC operating system was Windows 11, with PyTorch framework version 2.6.0 and CUDA architecture 12.4. Table 1 summarizes and lists the specific experimental environment.

To guarantee the fairness and robustness of the experiments, consistent parameter settings were implemented across all models. Input images were uniformly sized at 800 × 800 pixels, with training conducted for 100 epochs using a batch size of 16. The training was conducted using the SGD optimizer, configured with an initial learning rate (lr0) of 0.01, a learning rate decay factor (lrf) of 0.01 (final learning rate is lr0 × lrf = 0.0001), a momentum of 0.937, and a weight decay of 0.0005. Notably, the default mosaic data augmentation was disabled when training YOLO series models.

However, it should be emphasized that due to fundamental differences in algorithmic principles and network structures, certain models could not directly adapt to the aforementioned unified parameter settings. For these models, multiple sets of hyperparameter comparison experiments were conducted, and the optimal performance results were selected as the final comparison benchmark to ensure objectivity and scientific validity throughout the evaluation process. The specific details can be found in Table 2.

### 2.3. Evaluation Metrics

This research employs the precision, recall, 
${AP}_{50}$
, and 
AP
 metrics to systematically evaluate model performance in ship recognition.

Intersection over Union (*IoU*) is a widely used metric for object detection, measuring the overlap between predicted and ground truth boxes. It is defined as the ratio of their intersection to their union, as shown below:
(1)
$$IoU=\frac{Intersection\;Area}{Union\;Area}$$


In ship detection tasks, models may misclassify background regions and ship targets. The detection results can be categorized into four types: *TP*, *TN*, *FP*, and *FN*. Precision and recall are then defined based on these metrics, as shown in Equations (2) and (3):
(2)
$$P=\frac{TP}{TP+FP}\times 100\%,$$

(3)
$$R=\frac{TP}{TP+FN}\times 100\%$$


Average Precision (
AP
) is defined as the mean precision across different recall levels, computed as the area under the PR curve, where recall and precision are represented on the *x*- and *y*-axes, respectively (see Equation (4)):
(4)
$$AP=\int_{0}^{1}P(R)dR$$


MS COCO provides comprehensive evaluation metrics [47]. In this paper, 
${AP}_{50}$
 and 
AP
 are employed to assess ship detection performance. 
${AP}_{50}$
 represents the area under the PR curve at an IoU threshold of 0.5. 
AP
 (Average Precision) is a robust and comprehensive evaluation metric that measures precision across multiple IoU thresholds (0.5 to 0.95 in 0.05 increments), yielding an average of over 10 distinct levels.

### 2.4. The MC-ASFF-ShipYOLO Model

MC-ASFF-ShipYOLO is an improved model based on the YOLO11 object detection network. As a YOLO series model released on 30 September 2024 [43], YOLO11 achieves significant improvements in feature extraction capability and small object detection performance compared to previous versions, with the advantages of higher precision, fewer parameters, and faster inference speed.

This research proposes a dual optimization strategy for both backbone and head sections through an in-depth analysis of the YOLO11 network architecture. In the backbone section, we incorporate a Monte Carlo Attention mechanism to strengthen the network’s ability to capture the morphology and precise locations of small ship targets while also enhancing its understanding of contextual relationships between small objects and their surrounding environment, thereby effectively improving small object detection performance. In the head section, we add an Adaptive Spatial Feature Fusion (ASFF) module, enabling the model to more effectively consolidate multi-scale ship feature information, mitigating conflicts and inconsistencies between features at different scales, and significantly improving detection accuracy and robustness in complex multi-scale target scenarios. Figure 3 illustrates the overall architecture of the MC-ASFF-ShipYOLO model, while Figure 4 presents the detailed internal structure of each functional module.

#### 2.4.1. Monte Carlo Attention (MCAttn)

In CNN architectures, successive convolutional and pooling operations cause a progressive reduction in feature map resolution. For small ship target detection, these objects occupy extremely few pixels in images, limiting the available feature information for extraction. As network depth increases, these sparse features are easily compressed or even lost, leading to significant missed detections of small targets and consequently degrading the accuracy of the model’s overall detection. Therefore, enhancing the network’s learning capability for small target features and improving the model’s perception of contextual relationships between small objects and their surrounding environment are critical for detection performance improvement.

The proposed incorporation of Monte Carlo Attention into the YOLO network backbone effectively aids in small-ship target detection. MCAttn (Monte Carlo Attention) is a channel attention mechanism [48] that differs from the SE (squeeze–excitation) attention strategy [49], which obtains attention maps for each channel through global average pooling. Instead, MCAttn generates scale-invariant attention maps for each channel using pooling operations based on random sampling. Figure 5 illustrates the architecture of the MCAttn module.

MCAttn performs three average pooling operations on the input feature map to generate Pooled Tensors of different scales (3 × 3, 2 × 2, and 1 × 1). Then, from these three scales of Pooled Tensors, it randomly selects one 1 × 1 pooled tensor as the final attention map. MCAttn captures information at different scales, and this cross-scale characteristic deepens the model’s understanding of the relationship between small targets and their surrounding environment, enabling the model to better focus on the location of small ship targets, thereby improving recognition accuracy. The calculation steps for the MCAttn output attention map are as follows:
(5)
$$A_{m}\left(x\right)=\sum_{i=1}^{n}P(x,i)f(x,i),$$

(6)
$$\sum_{i=1}^{n}P\left(x,i\right)=1,$$

(7)
$$\prod_{i=1}^{n}P\left(x,i\right)=0$$


The MCAttn output attention map is defined as 
$A_{m}(x)$
, where *x* is the input tensor, *i* represents the size of the output attention map, 
f(x,i)
 represents the average pooling function, and the associated probability 
P(x,i)
 satisfies the constraints in Equations (6) and (7) above. *n* represents the number of pooling tensors.

Attention mechanisms commonly used in deep learning, like squeeze-and-excitation (SE), generate fixed-dimension attention maps by producing 1 × 1 output tensors through global average pooling (GAP), which often fails to function effectively in ship recognition tasks. Furthermore, this approach limits the model’s ability to capture cross-scale correlations, particularly when establishing non-linear relationships between small targets and their surrounding environments. MCAttn overcomes the limitations of conventional attention mechanisms by randomly sampling pooling tensors at various scales, significantly enhancing the model’s sensitivity to feature variations of small targets across multi-scale feature maps, thereby improving ship recognition accuracy.

#### 2.4.2. Adaptively Spatial Feature Fusion (ASFF)

As network depth increases, target detail features diminish while semantic information strengthens. Semantic features of small-scale targets can be fully extracted in shallow network layers, whereas large-scale targets require deeper network processing. This disparity may cause small target information to be lost during deep processing, making it difficult for models to achieve optimal detection balance. Furthermore, traditional multi-scale feature fusion networks (such as FPN) possess inherent deficiencies: when a target is classified as a positive sample in one feature layer, the corresponding region in other feature layers may be treated as background, resulting in feature conflicts and gradient computation interference. This inconsistency also exists in YOLO series networks. Therefore, we incorporated ASFF [50] into the detection head. ASFF adaptively learns spatial fusion weights for feature maps of different scales, effectively filtering conflicting information during training, enabling the model to utilize multi-scale features more efficiently, and enhancing feature scale invariance. The Detact + ASFF structure is illustrated in Figure 6, with the specific implementation requiring two steps: Feature Resizing and Adaptive Fusion.

Feature Resizing

The Neck component of YOLO11 generates three feature maps at different scales, namely P3, P4, and P5, each with distinct resolution and channel dimensions. Consequently, features from other levels require resizing to align with the features of the current level. For notational convenience, P3, P4, and P5 correspond to 
$Level\;l\;\left(l=1,2,3\right)$
. 
$x^{l}$
 is defined as the feature representation at resolution 
$Level\;l\;(l\in \left\{1,2,3\right\})$
. For any given 
Level l
, the feature maps 
$x^{n}$
 from other 
Levels n (n≠l)
 must be resized to match the dimensions of 
$x^{l}$
. We denote 
$x^{n\to l}$
 as the process of adjusting the feature map of 
$Level\;n\;\left(n\in \left\{1,2,3\right\}\right)$
 to conform to the dimensions of 
Level l
. This dimensional transformation is accomplished through appropriate up-sampling and down-sampling strategies.

2.Adaptive Fusion

After Feature Resizing, Adaptive Fusion is performed on features of three different scales. The calculation formula is as follows, where 
$x_{ij}^{n\to l}$
 denotes the feature vector at position 
(i,j)
 in the feature map from 
Level n
 after being resized to 
Level l
, and 
$y_{ij}^{l}$
 denotes the 
(i,j)
-th vector along the channel dimension in the output feature map 
$y^{l}$
.
(8)
$$y_{ij}^{l}=\alpha_{ij}^{l}\cdot x_{ij}^{1\to l}+\beta_{ij}^{l}\cdot x_{ij}^{2\to l}+\gamma_{ij}^{l}\cdot x_{ij}^{3\to l}$$



$\alpha_{ij}^{l},\;{\;\beta }_{ij}^{l},$
 and 
$\gamma_{ij}^{l}$
 represent the spatial importance weights of feature maps from 
$Level\;n\;\left(n\in \left\{1,2,3\right\}\right)$
 to 
Level l
, which are adaptively learned by the network. These weights can be simple scalars shared across all channels. They satisfy the constraints 
$\alpha_{ij}^{l}+\beta_{ij}^{l}+\gamma_{ij}^{l}=1$
 and 
$\alpha_{ij}^{l},\;\beta_{ij}^{l},\;\gamma_{ij}^{l}\in \left[0,1\right]$
. Taking 
$\alpha_{ij}^{l}$
 as an example, the calculation formula is as follows:
(9)
$$\alpha_{ij}^{l}=\frac{e^{\lambda_{\alpha_{ij}}^{l}\;}}{e^{\lambda_{\alpha_{ij}}^{l}\;}+e^{\lambda_{\beta_{ij}}^{l}\;}+e^{\lambda_{\gamma_{ij}}^{l}\;}}$$


Weight scalar mappings 
$\lambda_{\alpha }^{l},\;\lambda_{\beta }^{l},$
 and 
$\lambda_{\gamma }^{l}$
 are computed from 
$x^{1\to l},\;x^{2\to l},$
 and 
$x^{3\to l}$
, respectively, utilizing 1 × 1 convolution layers. Subsequently, 
$\alpha_{ij}^{l},\;\beta_{ij}^{l},$
 and 
$\gamma_{ij}^{l}$
 are defined through the Softmax function, with 
$\lambda_{\alpha_{ij}}^{l},\;\lambda_{\beta_{ij}}^{l},$
 and 
$\lambda_{\gamma_{ij}}^{l}$
 serving as control parameters. Through the ASFF module, MC-ASFF-ShipYOLO adaptively fuses spatial features across three scales, feeding the fusion results into detection heads to accomplish ship target detection.

#### 2.4.3. Loss Function

Confidence estimation, bounding box regression, and distribution focal loss jointly constitute the model’s loss function.

Considering the complex marine environment and the significant background noise affecting ship detection in SAR imagery, we adopt Binary Cross-Entropy (BCE) loss for confidence estimation, as it contributes to enhancing the model’s ability to distinguish foreground ships from complex backgrounds, thereby improving the accuracy of object existence prediction. For bounding box localization, the CIoU (Complete Intersection over Union) loss function is employed, demonstrating superior performance in high-precision box regression tasks. Moreover, compared to IoU, DIoU, and GIoU, CIoU is better suited for object detection scenarios with significant scale variations, making it an ideal choice for addressing the challenges of multi-scale ship detection.

Small objects occupy limited pixel regions in images, often accounting for less than 0.5% of the total image area (Section 2.1). This small size makes even minor deviations in bounding box predictions significantly impact localization accuracy. Distribution Focal Loss alleviates this challenge. The core concept of Distribution Focal Loss (DFL) is to predict bounding box coordinates using discrete probability distributions and optimize them through focal loss, concentrating predictions near the actual boundary values. Compared to traditional methods that directly predict bounding box coordinates, DFL offers greater flexibility and accuracy, enabling the model to capture subtle deviations and enhance adaptability for small-object detection [51]. Moreover, the focal mechanism also balances learning effectiveness across multi-scale objects. The calculation formula for DFL is presented below:
(10)
$$DFL\left(S_{i},S_{i+1}\right)=-\left(\left(y_{i+1}-y\right)\log{(S}_{i})+\left(y-y_{i}\right)\log\left(S_{i+1}\right)\right)$$


In the formula, 
$y_{i}$
 and 
$y_{i+1}$
 represent the closest discrete neighboring points to the ground truth value y, where 
$y_{i}=\left\lfloor y\right\rfloor$
 and 
$y_{i+1}=\left\lceil y\right\rceil$
. 
S(·)
 denotes the Softmax function, which generates probability distributions over discrete points, and 
$S_{i}$
 is the probability corresponding to 
$y_{i}$
 in the predicted distribution.

## 3. Results

This section provides a detailed evaluation of the MC-ASFF-ShipYOLO model’s performance on the constructed hybrid SAR ship dataset and systematically compares it with classical object detection networks. We conduct a comprehensive assessment of the improved model’s effectiveness in SAR ship detection tasks through both quantitative metrics and qualitative analysis. YOLO11 is an improved version of YOLOv8 [52]. Compared to the original model, YOLO11 introduces the Cross-Stage Partial with Self-Attention (C2PSA) module, which boosts the ability to capture contextual information, thereby improving detection performance for small-scale and occluded objects. Furthermore, the innovative C3k2 module replaces the traditional C2f module, optimizing both efficiency and inference speed while maintaining detection accuracy [44]. These advancements render YOLO11 particularly suitable for SAR ship detection. The YOLO11 architecture encompasses five model variants of varying scales (n, s, m, l, x), designed to address the diverse requirements of different dataset sizes and application scenarios. Based on these considerations, we selected YOLO11s as the improvement baseline.

### 3.1. Comparison Experiment

In this section, we chose the two-stage detectors, alongside the high-efficiency neural network model EfficientNet, and incorporated various editions of the YOLO series (such as YOLOv8 [52], YOLOv9 [53], YOLOv10 [54], YOLO11 [43], and YOLOv12 [55]) to conduct comparative experiments. Table 3 provides a detailed comparison of the experimental outcomes.

The analysis of the experimental data in Table 3 indicates that our selected baseline model, YOLO11s, demonstrates excellent performance on the hybrid SAR ship dataset, achieving 92.48% precision, 84.21% recall, 92.78% 
${AP}_{50}$
, and 67.40% 
AP
. Notably, YOLOv9s also performs well, with recall and 
AP
 values 0.73% and 0.13% higher than YOLO11s, respectively; however, its precision and 
${AP}_{50}$
 are 1.58% and 0.81% lower. Comprehensively evaluating detection performance and computational efficiency, YOLO11s emerges as the ideal base architecture for this research, offering faster inference speed (increased by approximately 37.04 FPS) compared to YOLOv9s.

The MC-ASFF-ShipYOLO proposed in this study significantly outperforms traditional algorithms including the YOLO series and Faster R-CNN in ship detection tasks under complex marine environments. The predefined anchor mechanism of Faster R-CNN exhibits reduced matching efficiency when facing multi-scale targets, resulting in decreased detection performance for small ships. Although Cascade R-CNN improves multi-scale adaptability through progressive anchor optimization, its general accuracy still requires improvement. In contrast, YOLO series models demonstrate superior general performance.

Our MC-ASFF-ShipYOLO model achieves significant performance improvements, reaching optimal levels in precision (93.87%), 
${AP}_{50}$
 (94.56%), and 
AP
 (70.44%), with recall at 86.84%, second only to EfficientNet. In-depth analysis reveals that while EfficientNet demonstrates the highest recall (89.29%), its precision is merely 74.91%, the lowest among all comparison models, indicating strong generalization ability and sensitivity to small object detection, but weak bounding box localization accuracy and complex background suppression capabilities, resulting in high recall but low precision with numerous false positives. Compared to the baseline model, MC-ASFF-ShipYOLO improves recall by 2.63%, exceeding YOLOv9s by 1.90%. The performance improvements are primarily attributed to the significant contributions of the MCAttn and ASFF modules in tackling the challenges of small object detection in complex environments and multi-scale ship recognition. The improved model exhibits greater stability during training without signs of overfitting.

### 3.2. Qualitative Analysis of Ship Detection Results

In this section, we perform ship target inference using the trained models and visualize the results in Figure 7. We selected the best-performing backbone from various object detection algorithms or the optimal size from the YOLO series to highlight the advanced nature of our improved model, MC-ASFF-ShipYOLO. For the parameter configuration, we used a confidence threshold of 0.75 and applied NMS with an IoU setting of 0.6. This selection is based on sufficient confidence in MC-ASFF-ShipYOLO, namely that the model can detect more ship targets even with higher confidence requirements while effectively reducing interference from low-quality detection results.

The visualization analysis of inference results in Figure 7 demonstrates that the improved MC-ASFF-ShipYOLO model can detect more ship targets, performing excellently even in complex environments or dense areas. This indicates that while enhancing attention to small ship targets, the model effectively suppresses noise interference from surrounding complex environments, exhibiting stronger generalization capability and robustness. Furthermore, these results confirm that MC-ASFF-ShipYOLO appropriately utilizes cross-scale information, enhancing its learning capacity for ship target features. In EfficientNet’s inference results, we observed fewer detected ships. This is due to the high confidence threshold (confidence threshold = 0.75), which filters out low-confidence targets, consistent with EfficientNet’s characteristics of high recall but low precision on the ship detection dataset.

## 4. Discussion

This section discusses the optimal position for introducing MCAttn into the backbone network, assessing the impact of the improved module through visualization-based qualitative analysis. Additionally, this section presents the comparison results of our improved module with similar existing methods, as well as the ablation experiment results.

### 4.1. Effectiveness of the MCAttn and Its Optimal Placement

Different features are extracted from various parts of the YOLO11 network backbone, necessitating a decision regarding which feature sections should utilize the MCAttn mechanism to maximize ship detection performance. We conducted a comparative experiment by introducing MCAttn after each C3k2 module in the backbone, with position illustrations shown in Figure 8. Table 4 lists the experimental results.

The experimental results revealed the critical impact of MCAttn module placement selection. Not all positions in the backbone where MCAttn was inserted could improve detection performance. When MCAttn was inserted after the P2 level C3k2 module (Posi-1), the model’s precision decreased by 1.95% and 
AP
 decreased by 0.74%, despite slight improvements in recall and 
${AP}_{50}$
 (by 0.23% and 0.16%, respectively). These minimal gains could hardly offset the precision loss. The appropriate placement of MCAttn is crucial for fully leveraging its cross-scale information capture capabilities. When MCAttn was positioned after the P5 level C3k2 module (Posi-4), all metrics achieved significant improvements: precision reached 92.62% (+0.14%), recall reached 85.56% (+1.35%), 
${AP}_{50}$
 reached 93.46% (+1.18%), and 
AP
 reached 69.34% (+1.94%). In particular, the substantial improvement in 
AP
 indicates significantly enhanced comprehensive detection capability, improved adaptability to multi-scale objects, and higher quality of bounding box predictions. Considering that 
AP
 is the most challenging evaluation metric, and given that all other metrics also improved, Posi-4 was determined to be the optimal insertion position for MCAttn. Notably, although the Posi-2 experiment achieved the highest recall (86.42%), its precision was only 90.55%, decreasing by 1.93%, indicating that the model may be overly sensitive to background noise and prone to false detections. This imbalance between precision and recall reflects diminished discriminative capability between background and target regions, which is disadvantageous for practical applications.

### 4.2. Comparative Evaluation of Similar Approaches

This section presents a comparative analysis with analogous methodologies to quantitatively assess the advantages of incorporating MCAttn and ASFF modules into the baseline model. For attention mechanisms, we selected CBAM [56] and SimAM [57] as comparative approaches and integrated them at identical positions as MCAttn. Regarding feature fusion, BiFPN [58] and HS-FPN [59] were implemented as comparative frameworks. Table 5 illustrates the experimental outcomes.

The experimental results indicate that various modules exert differential effects on YOLO11 performance. Regarding attention mechanisms, the 3D attention mechanism SimAM caused significant deterioration in model detection accuracy, with recall, 
$AP_{50}$
, and 
AP
 decreasing by 10.01%, 10.02%, and 10.28%, respectively. Channel attention mechanisms CBAM and MCAttn both enhanced model precision, with MCAttn demonstrating superior performance. Specifically, MCAttn outperformed CBAM by 0.07% and 0.49% in the precision and recall metrics, respectively, while exhibiting more substantial advantages in 
$AP_{50}$
 and 
AP
 metrics, surpassing CBAM by 0.14% and 1.07%, respectively. In terms of feature fusion modules, although the Multi-level Feature Fusion Pyramid (HS-FPN) and Bidirectional Feature Pyramid Network (BiFPN) significantly enhanced inference speed (increasing FPS by 129.63 and 155.95, respectively), both led to a decline in overall detection performance, with the more challenging 
AP
 metric decreasing by 5.46% and 0.90%, respectively. In comparison, the ASFF module achieved significant improvements in detection performance, increasing precision and recall by 0.75% and 0.41% and enhancing 
$AP_{50}$
 and 
AP
 by 0.82% and 0.99%, respectively. Consequently, integrating the MCAttn and ASFF modules into the YOLO11 architecture demonstrates greater advantages in improving ship detection performance.

### 4.3. Ablation Experiment

To systematically assess the impact of the proposed enhancement strategies on ship detection performance, a comprehensive set of ablation experiments was performed. Using YOLO11s as the baseline model, experiments were performed on the constructed hybrid SAR ship detection dataset, with detailed results presented in Table 6. By incrementally adding each functional module and analyzing the resulting performance changes, we were able to explicitly measure the individual contribution of each improvement as well as their combined effects, thereby confirming the effectiveness of the proposed approach.

The ablation experiment results in Table 6 demonstrate that the baseline YOLO11s model (Experiment 1) achieved 92.48% precision, 84.21% recall, 92.28% 
$AP_{50}$
, and 67.40% 
AP
 on the hybrid dataset, serving as the comparative benchmark for subsequent improvements. Upon incorporating MCAttn after the P5 level C3k2 module in the backbone (Experiment 2), all performance metrics improved. This comprehensive enhancement confirmed the effectiveness of the MCAttn module in strengthening feature learning, capturing cross-scale information, and increasing attention to small-ship targets. Adding the ASFF module to the detection head (Experiment 3) also yielded notable improvements: compared to the baseline model, precision increased by 0.75%, recall by 0.41%, 
$AP_{50}$
 by 0.82%, and 
AP
 by 0.99%. This indicates that ASFF effectively enhanced feature scale invariance through the adaptive learning of spatial fusion weights for feature maps at different scales, thereby improving ship detection accuracy.

In conclusion, the MC-ASFF-ShipYOLO model significantly enhanced detection accuracy for small ship targets in complex maritime scenarios while effectively addressing challenges in multi-scale feature extraction. Although detection performance has been enhanced, it is important to note that computational complexity increased correspondingly. Table 3 shows that the improved model has increased in parameter count, and the FPS has decreased to 232.56 images per second, presenting a trade-off that warrants further optimization in future work.

### 4.4. Analyze the Contribution of the Improvement Module

Ablation experiment results indicate that introducing the MCAttn module alone provided more significant overall accuracy improvements compared to the ASFF module. Relative to the ASFF experimental group, the MCAttn experimental group showed a 0.61% decrease in precision but achieved increases of 0.94% in recall, 0.36% in 
$AP_{50}$
, and 0.95% in 
AP
. To further analyze the contribution of the MCAttn module to ship detection, we visualized network output feature maps using Grad-CAM++ [60]. To ensure fairness in comparative experiments, all tests were conducted under identical configuration environments with the same parameters (e.g., Confidence = 0.75). Figure 9 displays the feature maps following the SPPF layer output in the backbone.

By comparing the feature heat maps, we can visually observe that after introducing the MCAttn module, the network’s attention to critical features of small ship targets significantly increased, with the fifth comparison group also demonstrating enhanced multi-scale detection capability. These changes are reflected in the heat maps, where ship regions exhibit higher activation intensity and more precise target localization [60]. The comparative results conclusively validate the positive contribution of the improved module in enhancing ship detection precision.

## 5. Conclusions

This study presented MC-ASFF-ShipYOLO, an improved framework for SAR ship detection, by introducing MCAttn and ASFF to enhance the YOLO model, which effectively mitigates the dual challenges of small target recognition and multi-scale feature integration. Through comprehensive experimentation on our constructed benchmark combining HRSID and SSDD datasets, we demonstrated the effectiveness of our two key innovations. Ablation experiment results demonstrate that both improvements enhanced model performance, with more significant effects when working synergistically. Comparative experiments with other widely used object detection algorithms confirm that MC-ASFF-ShipYOLO exhibits distinct advantages in small target recognition and multi-scale ship detection. Even at high confidence thresholds, the model generates high-quality detection results in dense and complex marine environments. We conducted an in-depth investigation of the optimal positioning for the MCAttn module and analyzed its positive impact on network feature activation using Grad-CAM++ visualization techniques. This research not only provides an effective implementation scheme for the YOLO11 model in SAR ship detection tasks but also establishes a novel technical approach for addressing the challenges of small ship identification and multi-scale ship detection in SAR imagery. The findings have practical significance for advancing the field of marine target monitoring.

Despite the progress achieved in this study, where our improved model demonstrated advantages in small ship detection and multi-scale challenges, certain limitations remain. The improved network architecture has increased computational overhead. Future research will focus on the following directions: We will explore more lightweight modules and optimization strategies. Additionally, we plan to implement rotated bounding box (PBox) annotation to better characterize ship geometrical features and reduce interference between adjacent targets in dense scenarios, further enhancing detection performance. These improvements will significantly enhance the model’s practical utility and applicability in complex marine environments.

## Figures and Tables

**Figure 1 sensors-25-02940-f001:**
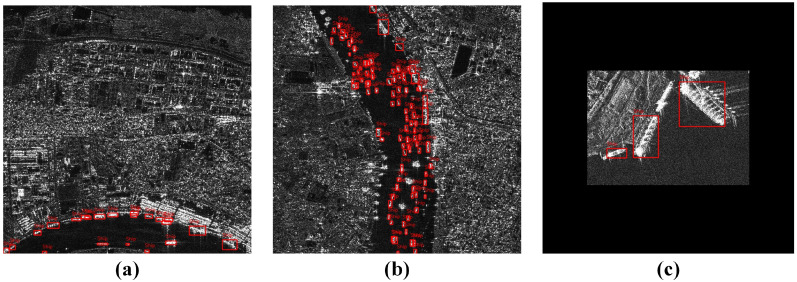
(**a**) Ship targets in complex inland river environments; (**b**) scene showing densely distributed small ship targets; (**c**) examples of multi-scale ship targets after zero-padding processing of the SSDD dataset. Note that (**a**,**b**) are from the HRSID dataset. Red bounding boxes highlight real ships.

**Figure 2 sensors-25-02940-f002:**
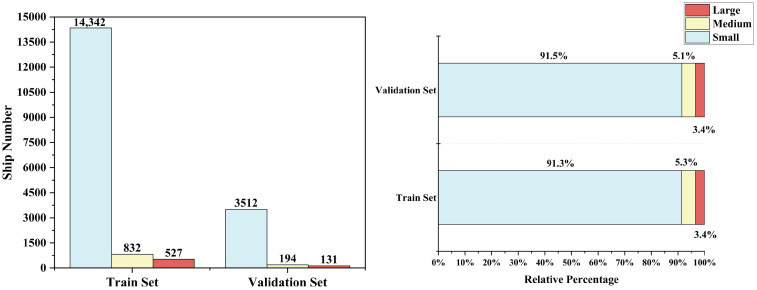
Number of ships of different scales and the proportion of each scale in the training and validation sets.

**Figure 3 sensors-25-02940-f003:**
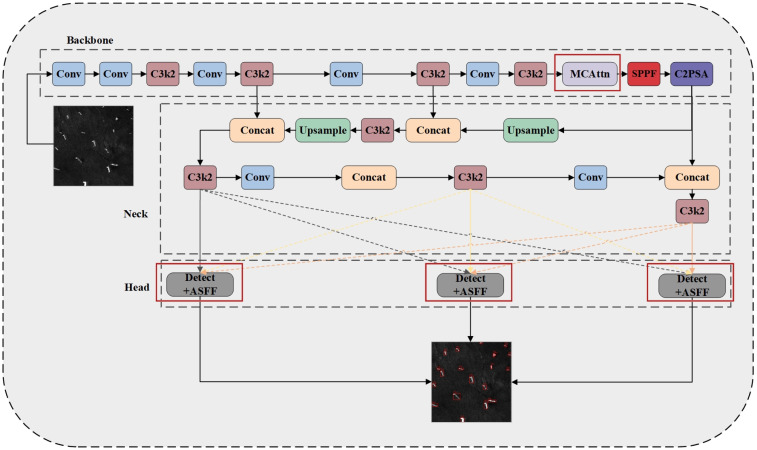
Overall framework of the MC-ASFF-ShipYOLO model based on YOLO11 improvements, with improved modules highlighted in red boxes. The foundational YOLO11 network structure is independently drawn based on the Ultralytics configuration files [43].

**Figure 4 sensors-25-02940-f004:**
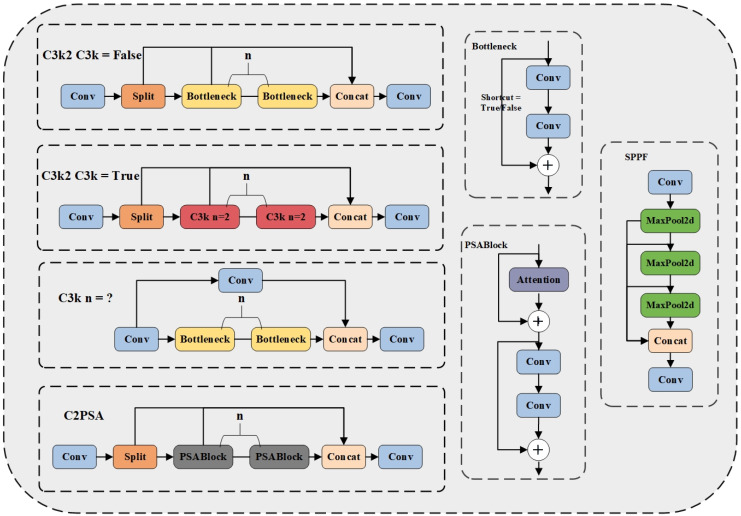
Detailed composition of modules in the improved MC-ASFF-ShipYOLO model.

**Figure 5 sensors-25-02940-f005:**
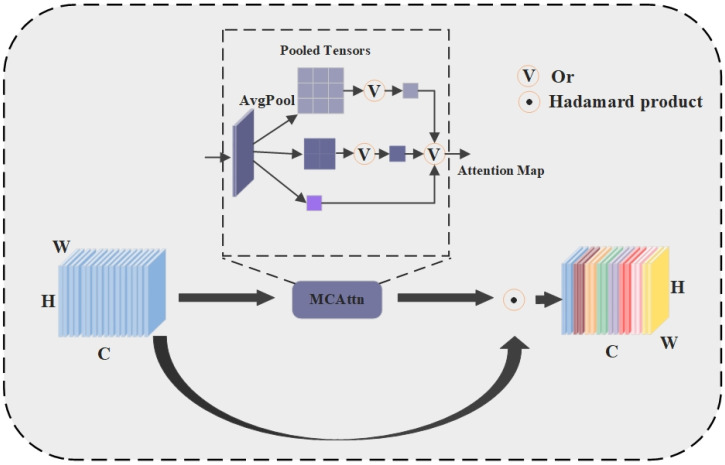
Detailed implementation structure of Monte Carlo Attention. The letters H, W, and C denote Height, Width, and Channels respectively.

**Figure 6 sensors-25-02940-f006:**
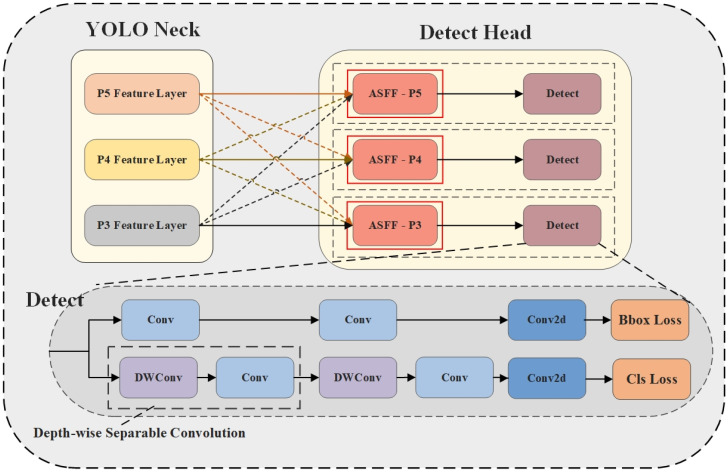
Add detailed implementation diagram of ASFF to the head section. The P3, P4, and P5 feature layers are output by the Neck component of YOLO. YOLO incorporates three detection heads, each dedicated to detecting objects at different scales. The Adaptive Spatial Feature Fusion (ASFF) module is integrated immediately preceding each detection head.

**Figure 7 sensors-25-02940-f007:**
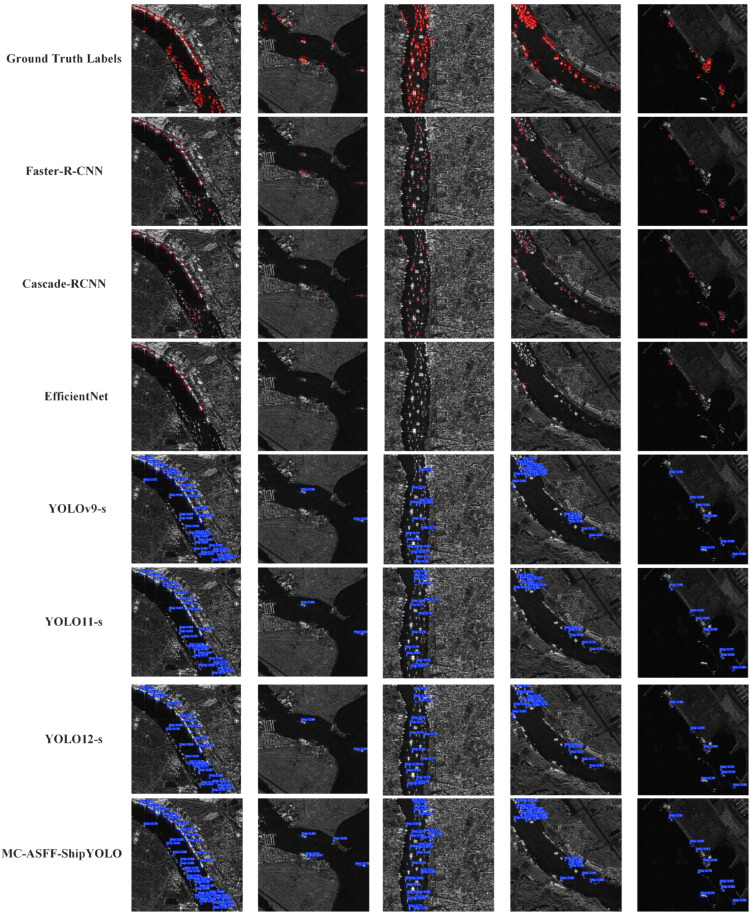
The MC-ASFF-ShipYOLO model is compared with the YOLO series models and traditional object detection models. Both Faster-RCNN and Cascade-RCNN use ResNet50 as the backbone.

**Figure 8 sensors-25-02940-f008:**
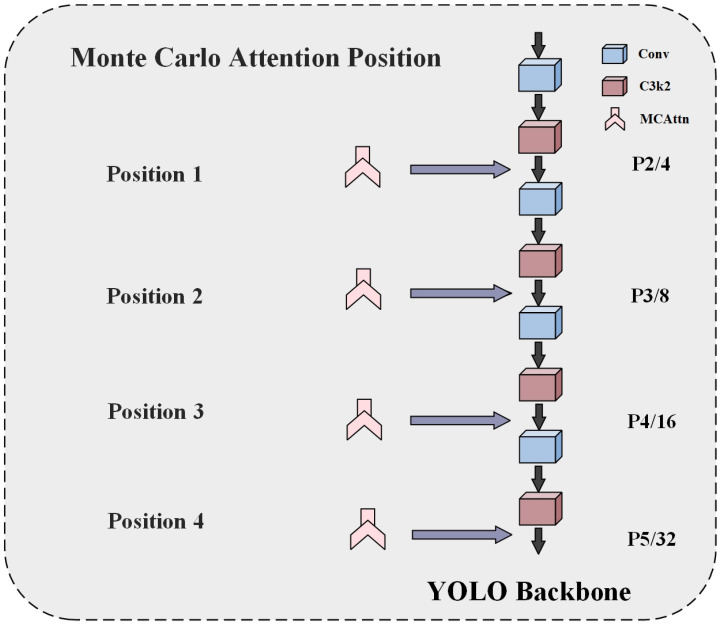
Alternative positions for MCAttn module insertion in the YOLO11 backbone. PX/Y: feature map at pyramid level X (X = 2,3,4,5), Y represents the down-sampling factor. In the figure, position i (i = 1, 2, 3, 4) corresponds to the four experimental groups (Posi-i) presented in Table 4.

**Figure 9 sensors-25-02940-f009:**
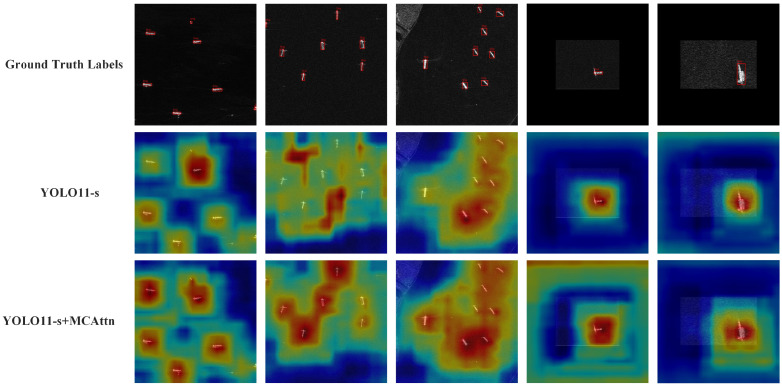
Visualization results of the Grad-CAM++ feature heat map for ship target recognition (Confidence = 0.75). The feature heat maps of YOLO11-s and YOLO11-s + MCAttn are both outputs after the backbone SPPF layer.

**Table 1 sensors-25-02940-t001:** Experimental environment.

Configuration	Model/Parameter
Operation system	Windows 11
CPU	Intel(R) Xeon(R) Gold 6430 @2.10 GHz
GPU	NVIDIA GeForce RTX 4090 (24,564 MiB)
RAM	120 GB
Compiler	Python3.11.11
Framework	CUDA12.4/cudnn9.1.0/torch2.6.0

**Table 2 sensors-25-02940-t002:** Hyperparameter settings for other models. All models used the SGD optimizer (momentum = 0.937, weight decay = 0.0005). “-” in the NMS column indicates that non-maximum suppression was disabled during training. In the Backbone column, ‘R’ denotes ResNet and ‘Eff-b3’ denotes EfficientNet-B3.

Models	Backbone	Batch Size	lr0	lrf	NMS
Faster R-CNN	R50	16	0.01	0.0001	0.6
	R101	12	0.01	0.0001	0.7
Cascade R-CNN	R50	16	0.01	0.0001	0.6
	R101	16	0.01	0.0001	0.7
EfficientNet [46]	Eff-b3	6	0.01	0.0001	-

**Table 3 sensors-25-02940-t003:** Experimental environment. In the Backbone or Size column, ‘R’ denotes ResNet and ‘Eff-b3’ denotes EfficientNet-B3. P (Precision), R (Recall), 
$AP_{50}$
 (Average Precision at IoU threshold of 0.5), 
AP
 (AP averaged over IoU thresholds in [0.5:0.95] (step = 0.05)), and FPS (images per second, measuring inference speed).

Models	Backbone or Size	P /%	R /%	$AP_{50}$ /%	AP /%	Params (M)	FPS (img/s)
YOLOv8	n	90.85	82.83	90.67	64.05	3.01	666.67
	s	90.74	84.75	91.81	65.24	11.13	370.37
YOLOv9	t	91.11	83.82	91.55	65.76	1.97	555.57
	s	90.90	84.94	91.97	67.53	7.17	333.33
YOLOv10	n	88.90	81.55	90.11	65.20	2.70	666.67
	s	91.09	83.40	91.76	66.77	8.04	384.62
YOLO11	n	90.41	81.56	89.20	63.92	2.58	555.57
	s	92.48	84.21	92.78	67.40	9.41	370.37
YOLOv12	n	90.93	81.08	90.43	66.04	2.56	416.67
	s	91.39	82.21	91.59	66.24	9.23	243.90
EfficientNet	Eff-b3	74.91	89.29	88.00	61.30	18.34	306.30
Faster R-CNN	R50	84.18	82.87	83.40	60.80	41.35	308.10
	R101	84.18	83.08	83.20	60.50	60.34	324.00
Cascade R-CNN	R50	84.51	83.98	84.50	63.10	69.15	317.50
	R101	84.97	84.13	83.80	62.80	88.14	317.10
MS-ASFF- ShipYOLO (Ours)	-	93.87	86.84	94.56	70.44	60.28	232.56

**Table 4 sensors-25-02940-t004:** Effect of MCAttn insertion at different positions.

Models	P /%	R /%	$AP_{50}$ /%	AP /%
Baseline	92.48	84.21	92.28	67.40
Posi-1	90.53	84.44	92.44	66.66
Posi-2	90.55	86.42	93.20	68.28
Posi-3	92.37	85.28	93.45	69.17
Posi-4	92.62	85.56	93.46	69.34

**Table 5 sensors-25-02940-t005:** Comparison experiments of different modules (baseline: YOLO11s).

Model	P /%	R /%	$AP_{50}$ /%	AP /%	FPS (img/s)
Baseline	92.48	84.21	92.28	67.40	370.37
+SimAM	92.11	74.20	82.26	57.12	434.78
+CBAM	92.55	85.07	93.32	68.27	263.16
+MCAttn	92.62	85.56	93.46	69.34	270.27
+HS-FPN	91.31	82.20	89.56	61.94	500.00
+BiFPN	91.45	83.37	91.94	66.50	526.32
+ASFF	93.23	84.62	93.10	68.39	303.03

**Table 6 sensors-25-02940-t006:** Ablation experiment. “🗸” indicates the module is added, and “🗶” indicates it is not.

Different YOLO11s Models	MCAttn	ASFF	P /%	R /%	$AP_{50}$ /%	AP /%	FPS (img/s)
Experiment 1 (Baseline)	🗶	🗶	92.48	84.21	92.28	67.40	370.37
Experiment 2	🗸	🗶	92.62	85.56	93.46	69.34	270.27
Experiment 3	🗶	🗸	93.23	84.62	93.10	68.39	303.03
Experiment 4	**🗸**	**🗸**	93.87	86.84	94.56	70.44	232.56

## Data Availability

The data presented in this study are available on request from the corresponding author.

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
