# Peer review of "MC-ASFF-ShipYOLO: Improved Algorithm for Small-Target and Multi-Scale Ship Detection for Synthetic Aperture Radar (SAR) Images"

_sensors, 2025, doi:10.3390/s25092940_

Round 1

Reviewer 1 Report

Comments and Suggestions for Authors

In this manuscript, the authors propose an improved ship detection model named MC-ASFF-ShipYOLO. This model enhances small target detection by enhancing feature extraction across targets of different scales.
To achieve this they introduced the MCAttn module to enhance the backbone network performance of the YOLO 11 model and they incorporated the ASFF module into the detection head to improve multi-scale feature extraction.
Their work aims to provide a more effective solution for ship detection in SAR images.

I have some minor suggestions for the authors:

1. At line 214, the authors state that for some models the set of hyperparameters has been chosen by conducting experiments. It would be nice to add a table showing the hyperparameters used for each model mentioned.
2. Since the computational complexity of the models is mentioned more times in this manuscript the authors should add both the computational complexity, the inference speed (FPS or latency), and the training time to Table 2 (page 12).
3. At line 379, the authors state that they selected YOLO 11 as the baseline model after "comprehensive consideration". It is worth to describe these considerations in more detail.
4. The description of the loss function in 2.4.3 could be improved to better explain the design choices.
5. The authors should consider publishing the dataset described in section 2 to allow future research to use the dataset for a fair comparison.

Reviewer 2 Report

Comments and Suggestions for Authors

The authors of this manuscript propose MC-ASFF-ShipYOLO, a framework addressing both small target recognition and multi-scale ship detection.

This article needs a major revision, the authors need to address the following issues:

  • In the abstract, the authors state, “Relatively few studies have comprehensively addressed both small target identification difficulties and multi-scale ship detection simultaneously.” However, there is already a substantial body of research on small target identification and multi-scale object detection. The authors are requested to verify the objectivity of this statement and revise it to accurately reflect the existing literature.
  • The novelty of the proposed method is not clearly articulated. Both the MCAttn and ASFF modules in the proposed method are adopted from existing work. It is recommended that the authors provide a comparative analysis with similar methods (e.g., CBAM, CoAM, BiFPN, SEPC) to explicitly highlight the unique advantages or improvements of MCAttn and ASFF.
  • The paper mentions that the proposed model improves performance at the cost of increased computational complexity but does not provide specific data to support this claim. It is suggested that the authors include a quantitative comparison of MC-ASFF-ShipYOLO with baseline models in terms of FLOPs, parameter count, and inference time to clearly demonstrate the computational overhead.
  • Some tables (e.g., Table 2) do not define the metrics used (e.g., P, R, APâ‚…â‚€) in the table captions or footnotes. It is recommended that these definitions be added for clarity. Additionally, the authors are encouraged to enhance the visual presentation of figures (e.g., Figures 6 and 8) and include appropriate legends or annotations to improve readability and comprehension.
  • The authors have addressed small target and multi-scale ship detection tailored to the characteristics of SAR images. There are numerous existing studies focusing on SAR image characteristics, such as the following references, which the authors are encouraged to cite and discuss: “Hierarchical and progressive learning with key point sensitive loss for sonar image classification” and “MSSD-Net: Multi-Scale SAR Ship Detection Network.” The authors should clarify how their proposed method differs from or improves upon these existing approaches.

Comments on the Quality of English Language

The English could be improved to more clearly express the research.

Round 2

Reviewer 2 Report

Comments and Suggestions for Authors

All my concerns have been solved and I recommend publishing this work.